# **OpenLandMap-soildb: global soil information at 30 m spatial resolution for 2000–2022+ based on spatiotemporal Machine Learning and harmonized legacy soil samples and observations**

Tomislav Hengl<sup>1</sup>, Davide Consoli<sup>1</sup>, Xuemeng Tian<sup>1</sup>, Travis W. Nauman<sup>2</sup>, Madlene Nussbaum<sup>3</sup>, Mustafa Serkan Isik<sup>1</sup>, Leandro Parente<sup>1</sup>, Yu-Feng Ho<sup>1</sup>, Rolf Simoes<sup>1</sup>, Surya Gupta<sup>4</sup>, Alessandro Samuel-Rosa<sup>5</sup>, Taciara Zborowski Horst<sup>6</sup>, José L. Safanelli<sup>7</sup>, and Nancy Harris<sup>8</sup>

<sup>1</sup>OpenGeoHub Foundation, Doorwerth, The Netherlands

<sup>3</sup>University of Utrecht, Utrecht, the Netherlands

<sup>4</sup>Department of Environmental Sciences, University of Basel, Basel 4056, Switzerland

<sup>5</sup>Universidade Tecnológica Federal do Paraná, Santa Helena, Paraná, Brazil

<sup>6</sup>Universidade Tecnológica Federal do Paraná, Dois Vizinhos, Paraná, Brazil

<sup>7</sup>Woodwell Climate Research Center, Falmouth, MA, USA

<sup>8</sup>World Resources Institute, Washington DC, USA

Correspondence: Davide Consoli (davide.consoli@opengeohub.org)

**Abstract.** There is increasing interest in global dynamic soil information with changes in soil properties mapped over time and at high spatial resolution. Thanks to long-term, multi-temporal, and fine- and medium-resolution satellite missions such as Landsat, MODIS, Copernicus Sentinel and similar, it is possible to produce globally consistent predictions of key soil variables that match other 10–30 m spatial resolution global data sets. This paper describes data preparation, modeling, and production of

- OpenLandMap-soildb: global dynamic predictions of soil organic carbon content, soil organic carbon density, bulk density, soil pH in  $H_2O$ , soil texture fractions (clay, sand and slit) and USDA subgroup soil types (USDA soil taxonomy subgroups) at 30 m spatial resolution based on spatiotemporal Machine Learning (Quantile Regression Random Forest with output predictions showing the mean plus the lower and upper prediction intervals of 68% probability). To train the models, a large compilation of soil samples imported from legacy soil projects was used: 216,000 soil samples with soil carbon density (kg m<sup>-3</sup>), 408,000
- soil samples with soil carbon content (g kg<sup>-1</sup>), 272,000 samples with soil pH in H<sub>2</sub>O, 363,000 samples with clay, silt, and sand (%), and 134,000 samples with bulk density oven dry (t m<sup>-3</sup>). Soil carbon and soil pH were mapped with 5-year time-intervals; soil texture fractions, bulk density, and soil types were mapped for recent years only. The cross-validation results indicate RMSE of 17.7 (kg m<sup>-3</sup>; 0.486 in log-scale) and CCC of 0.88 for SOC density, RMSE of 51.3 (g kg<sup>-1</sup>; 0.574 in log-scale) and CCC of 0.87 for SOC content, RMSE of 0.15 (t m<sup>-3</sup>) and CCC of 0.92 for bulk density of fine-earth, RMSE
- of 0.51 and CCC of 0.91 for soil pH, RMSE of 8.4% and CCC of 0.87 for soil clay content, and RMSE of 12.6% and CCC of 0.84 for soil sand content respectively. The most important variables for predicting soil organic carbon density (kg m<sup>-3</sup>) were: soil depth, Landsat-based uncalibrated Gross Primary Productivity (GPP), Normalized Difference Vegetation Index (NDVI) and CHELSA bioclimatic indices. The global distribution of soil pH can be primarily explained by the CHELSA Aridity Index (long-term), annual precipitation, and salinity grade. The global stocks for 2020–2022+ period for 0–30 cm depth interval are

<sup>&</sup>lt;sup>2</sup>Moab, UT, USA

estimated at 461 Pg (Peta grams); the results further indicate that, in the last 25 years, the world has lost at least 11 Pg of SOC in the top soil. Suggestions are made on how to set up global permanent monitoring stations to accurately track land degradation and enable land restoration projects. The training dataset is available at https://doi.org/10.5281/zenodo.4748499 (Hengl and Gupta, 2025), while the resulting data products can be accessed at https://doi.org/10.5281/zenodo.15470431 (Consoli et al., 2025). Both datasets are released under a *CC-BY* license.

#### 1 Introduction

Soils symbolize fertility and are the foundation of our civilization; one of the most undervalued natural resources. Changing that perspective is a mission worth dedicating a career. Common modern threats to soil health include the loss of organic matter, the loss of biodiversity, soil pollution, soil salinization, and soil erosion. There is an increasing focus on soils due to

- their importance for ecosystem services: from growing crops, to filtering water, and providing building material (Smith et al., 2020). Soils are also one of the potential carbon pools that could significantly help decrease greenhouse gas (GHG) emissions in the atmosphere. Unsustainable land use and population pressure are the main drivers of soil degradation (Montgomery, 2007; Borrelli et al., 2017; Kraamwinkel et al., 2021). We are at a crossroads in history in our attempt to preserve soil resources before we completely lose them.
- It is, in fact, a striking paradox that on the one hand, soils are one of the most promising solutions for mitigating greenhouse gas emissions, while, on the other hand, 60–70% of soils are currently unhealthy (Panagos et al., 2022). In the last 150 years, half of the topsoil on the planet has been degraded due to erosion, compaction, desertification, acidification, and loss of organic carbon and primary nutrients; mostly due to changes in global land use and climate. Hou et al. (2025) estimate that 14–17% of all croplands are polluted with toxic metals exceeding agricultural thresholds. Moreover, soil erosion could increase up to 60%
- in the next 30 years (Borrelli et al., 2017). For instance, the Continental United States alone may lose 1.8 Pg (petagrams) of soil organic carbon under climate change (Gautam et al., 2022). Padarian et al. (2022a) estimates that agricultural land could lose approximately 14% of the carbon sequestration potential of soil by 2040 due to climate change. Meanwhile, some recent estimates by Sasmito et al. (2025) indicate that half of the land use carbon emissions in Southeast Asia can be mitigated through the peat swamp forest and mangrove conservation and restoration. Padarian et al. (2022a) estimates the additional SOC storage
- potential in the topsoil of global croplands to be between 29 to 65 Pg C.

The ability to measure and evaluate progress towards maintaining or restoring healthy soils will be critical to the success of improved land management promoted by stakeholders and policy makers. Today, every land manager should have easy access to verified GHG emissions and removal data at the parcel level, and carbon farming must support the achievement of the proposed net removal targets — for example, 310 Mt CO2eq in the land sector in the EU until 2030 (Searchinger et al.,

2022). However, the production of reliable estimates of global SOC stocks and SOC carbon sequestration has proven complex (Scharlemann et al., 2014; Minasny et al., 2017). The uncertainty in the estimates of the total organic carbon stocks in the soil of our planet for the 0–1 m depth interval is large (Scharlemann et al., 2014; Tifafi et al., 2018; Feeney et al., 2022; Lin et al., 2022), leading to problems of general credibility of these maps.

Direct measurement of soil properties from space is cumbersome (van Wesemael et al., 2024; Broeg et al., 2024; Li et al., 2024). Soils are often hidden below the surface under dense vegetation, and most EO systems do not penetrate the soil. Saha et al. (2024) reviewed the direct use of EO products and systems to monitor SOC from space and concluded that direct SOC detection is limited due to the low signal-to-noise ratio and low spectral resolution: most predictive mapping models have a limited  $R^2$  between 0.3 and 0.7. Even bare surface spectra can be used to represent only the first few centimeters of topsoil,

limited R<sup>2</sup> between 0.3 and 0.7. Even bare surface spectra can be used to represent only the first few centimeters of topsoil, while, on the other hand, many studies often ignore soil management practices such as crop rotation, conservation tillage practices, fertilization level, plow depth, addition of manure to soil, and similar (Saha et al., 2024).

The uncertainty about how much organic carbon is in the soil and how much could potentially be sequestered appears to be high, especially for northern latitudes, tropical peatlands / wetlands and semi-arid areas (Crowther et al., 2016; Lin et al.,

- 2022). The most up-to-date point data from Canada and the Russian Federation now indicate that large pools of soil organic matter in tundra and taiga-like biomes have probably been underestimated in previous global maps (Shaw et al., 2018; Wagner et al., 2023). Global warming and rising temperatures are likely to perpetuate the release of soil carbon in high-latitude areas dominated by permafrost (Crowther et al., 2016; Van Gestel et al., 2018). Therefore, accurate estimates of the carbon budget beyond 60° north, including the distribution of peatland soils (covering only 2–3% of the total area, but representing probably
- 40–50% of total stocks), are increasingly important. In tropical areas, Xu et al. (2018) and Gumbricht et al. (2017) have estimated that the extent of peatlands is somewhat larger than expected (currently estimated to be 2.8% of the total land mask), and there appear to be still many unmapped bogs of peat and organic material, especially in Latin America (Gumbricht et al., 2017), Africa (Fatoyinbo, 2017), and mangrove forests (Atwood et al., 2017). Deforestation and degradation of tropical forests appear to also perpetuate the loss of SOC (Drake et al., 2019).
- Some of the most recent global maps of SOC at 1 km and 250 m are provided by FAO (2022) and Poggio et al. (2021). At the continental level, Yigini and Panagos (2016) produced detailed SOC maps for Europe; Liang et al. (2019) for China; Hengl et al. (2021) for Africa; Guevara et al. (2018) for South America; Ramcharan et al. (2018) and Nauman et al. (2024) for the United States. Beyond mapping the general spatial distribution of SOC, there is also an increasing interest in mapping changes in soil properties over time, with a special focus on soil carbon, soil nitrogen, pH, and other soil nutrients that are
- more dynamic and prone to land management changes (National Academies of Sciences, Engineering, and Medicine and others, 2021; Broeg et al., 2024; Li et al., 2024). Although soils change gradually, often on a scale of a few hundred years, locally there can be drastic effects, especially as a result of land degradation or sudden change of land use. In general, current systems in place to monitor soil properties (physical, chemical, and biological characteristics) together with soil loss and soil degradation measures do not provide sufficient information to accurately quantify changes in soil resources over time (National
- Academies of Sciences, Engineering, and Medicine and others, 2021).

The three most common groups of soil properties of interest for dynamic mapping are soil organic carbon stocks, soil nutrients (Chen et al., 2022), and soil hydrological properties such as available soil water (López-Ballesteros et al., 2023) and soil moisture content. Guo and Gifford (2002); Stockmann et al. (2015), and Stumpf et al. (2018) focused on modeling changes in SOC primarily as an effect of changes in land use and/or land cover over decades. The second most important soil-forming

or controlling factor for predicting SOC changes at large scales is climate. Jones et al. (2005) and Gottschalk et al. (2012),

25

for example, provide estimates of changes in SOC due to climate change, with a special focus on predicting potential SOC losses in the future. Padarian et al. (2022b) proposed a two-step semi-mechanistic framework to model SOC over time: first, the baseline of the SOC stock is estimated using predictive mapping (in this case the baseline is the year 2001), and second, the SOC values are then propagated year by year over time by incorporating changes in land cover. Padarian et al. (2022a) uses

- a similar data set to estimate the SOC sequestration potential for agricultural land. Heuvelink et al. (2021) mapped the SOC dynamics of Argentina at 250 m spatial resolution using a time series of NDVI images for 1982–2017 and Random Forest. Their results indicate that, in fact, bio-climatic variables are somewhat more important than NDVI images for modeling SOC. Ugbemuna Ugbaje et al. (2024) developed spacetime predictions of SOC stocks for Australia at a 90 m spatial resolution covering 1990 and 2018. Venter et al. (2021) produced three decades of predictions of top-soil stocks for South Africa at 30 m
- spatial resolution; based on the time-series of predictions authors also provide estimates of soil carbon change in kg m<sup>-2</sup> (for 0–30 cm depth interval). van Wesemael et al. (2024) produced triannual predictions (2018–2020, 2019–2021 and 2020–2022) of top-soil SOC (in %) for European Union, using a combination of spectral models for croplands (bare surface soil spectra) and the digital soil mapping approach for forest and grasslands.
- Currently, the most referenced global soil data set with prediction intervals per pixel is SoilGrids V2.0 available at 250 m spatial resolution (Poggio et al., 2021). In addition, the FAO has recently updated the Harmonized World Soil Database (HWSDV), produced at 1 km spatial resolution (FAO & IIASA, 2023) and is also maintaining the Global Soil Partnership's GSOCmap (FAO, 2022). In practice, all three (SoilGrids V2.0, GSOCmap, and HWSDB) are lagging behind in spatial resolution with comparable global vegetation data sets, now usually focusing at 30 m or even 10 m, e.g., representing land cover dynamics (Potapov et al., 2020), crop classification (Van Tricht et al., 2023), forest canopy parameters (Turubanova et al., 2023), and similar. In addition, updating global soil maps for shorter periods, such as 1–2 times a year, has never materialized.

In this paper, we describe a fully documented open framework for producing predictions of primary dynamic soil properties at 30 m spatial resolution for the period 2000–2022+ (5–year composites), in addition to the spatial distribution of soil types. We focus on the following four research questions:

- R1: Do Landsat 30 m resolution images help improve the accuracy of predictions? If so, which Landsat-derived biophysical indices are the key for soil mapping?
- R2: How well do predictions from global models compare to observed values at locations not used in the map calibration/training, i.e., what is the expected prediction error at unvisited locations?
- R3: What are the key drivers leading to changes in SOC? How, for example, does conversion of tropical forests to croplands and pasturelands impact SOC and pH on a scale of 25+ years?
- 30 R4: What are the world's remaining hotspots of SOC stocks?

We first present in detail all the data preparation, modeling, and prediction steps and how accuracy was assessed using robust procedures. In the results section, we report results of standardization, accuracy assessment, and change-analysis. We also provide visual evidence of patterns in the predictions and zoom in on the potential drivers of change in soil properties.

The data and code used to produce the results and instructions on how to access the data are publicly available through https://github.com/openlandmap/soildb.

# 2 Materials and methods

In the following sections, we explain in detail how the point (training) data were prepared, how the covariate layers were selected and prepared for analysis, how and why we inserted pseudo-observations, and why we have made some design choices. In addition, we explain how we conducted cross-validation and how the prediction intervals were derived (per pixel). We run extensive tests to check predictive performance and then report results in both original and transformed spaces, which is especially important for log-normal and composite variables.

# 2.1 Spatiotemporal Machine Learning

- We developed a fully automated global soil mapping framework based on a large stack of covariate layers representing the standard soil-forming and controlling factors (relief, climate, parent material, living ecosystem, and human impact) (Jenny, 1994) and an optimized machine learning pipeline as implemented in the scikit-map library for Python. The general soil mapping framework is illustrated in Fig. 1 and has been used to predict continuous dynamic soil variables and static soil properties, i.e., soil types and physical soil properties. We refer to the mapping framework as the "*EO-SoilMapper*" because
- the most important covariate data are the Earth Observation (EO) time series of images. We are able to produce predictions at 30 m and for a period of almost 25 years, mainly because we use the complete and cloud-free Landsat Archive previously prepared by Consoli et al. (2024), and the global digital terrain model (DTM) and its multiscale variables produced by Ho et al. (2025).

Spatio-temporal Machine Learning (ML) implies (Hackländer et al., 2024; Tian et al., 2024):

- 1. *Spatio-temporal overlay*: observations & measurements (O&M) are overlaid with covariate layers by matching both the geographic location and the start / end time period. In this paper, we only match observations by year, although some soil properties, such as soil moisture, would also require refined temporal identification.
  - 2. *Strictly defined time-period of interest*: covariate layers need to match the distribution of O&M's in the time domain, i.e., there needs to be enough training points spread across the period of interest (in this case 2000–2022+).
- 3. Spatio-temporal cross-validation: for accuracy assessment, we report both spatial blocking cross-validation and leaveone-year-out (LOYO) cross-validation to prevent producing over-optimistic validation results for densely sampled/clustered points, due to e.g. strong spatial auto-correlation.
  - 4. *Predictions in spacetime using spacetime blocks*: predictions are strictly spatio-temporal, i.e., they are connected with certain begin/end time periods. We refer to the spacetime prediction reference as *"spacetime blocks"*.