# Peer review of "OpenLandMap-soildb: global soil information at 30 m spatial resolution for 2000–2022+ based on spatiotemporal Machine Learning and harmonized legacy soil samples and observations"

_Earth System Science Data, 2025_

## Community Comment (CC1)

**Comments**

**Title**: *Open LandMap-soildb: Enabling High-Resolution Soil Intelligence for Climate, Land Restoration, and Agricultural Policy*

Soil degradation is a growing global crisis, threatening food security, carbon sequestration potential, and ecological resilience. To tackle this, precise, spatially explicit, and temporally consistent soil information is essential. The newly developed *OpenLandMap-soildb* offers an unprecedented advancement in this space, providing global soil data at a fine spatial resolution (30 m) across two decades (2000–2022), using a spatiotemporal machine learning framework and harmonized legacy datasets.

This initiative delivers dynamic predictions for key soil properties including soil organic carbon (SOC) content and density, bulk density, soil pH, and USDA soil types. These outputs are based on over 1 million quality-controlled and harmonized soil samples, combined with Earth Observation (EO) satellite data, terrain models, and climatic indicators. Notably, the study estimates that the planet has lost more than 11 petagrams (Pg) of SOC in the top 30 cm of soil over the last 25 years, a signal of worsening land degradation and a missed opportunity for carbon sequestration.

This manuscript presents an ambitious and technically compelling global soil dataset spanning over two decades at high spatial resolution. The integration of legacy soil samples with modern satellite-derived covariates via machine learning methods is a noteworthy advancement for soil science and spatial ecology. However, certain methodological and interpretative aspects warrant clarification and refinement before publication.

**Major Concerns**

1. **Model Transparency and Reproducibility**

   o The use of Quantile Regression Random Forests is appropriate, but the manuscript lacks sufficient detail regarding hyperparameter optimization, feature selection criteria, and potential overfitting mitigation strategies.

   o The approach to uncertainty quantification is promising; however, clearer guidance on interpreting prediction intervals in practical applications would enhance user comprehension.

2. **Temporal Granularity**

   o Five-year intervals may oversimplify dynamic changes due to land use transitions or climate events. The authors should discuss how these limitations affect the detection of soil change patterns.

3. **Spatial Validation Design**

- There is limited description of spatial cross-validation strategies. It's essential to confirm the use of geographically independent test sets to avoid inflating predictive performance due to spatial autocorrelation.

4. **Legacy Data Harmonization**

- While the dataset is impressively large, the harmonization process of legacy samples (e.g., sampling depths, analytical methods, and metadata consistency) needs greater transparency. Including a harmonization workflow or uncertainty estimates tied to legacy data variability would be beneficial.

5. **Spatial Data Bias**

- Over-representation of North America and Europe; sparse coverage in Asia, Russia, and Africa. This introduces spatial bias, which may influence the global model predictions unfairly, especially for underrepresented biomes and land-use systems.

6. **Model decision**

    Despite high accuracy, it reduces interpretability for policymakers or non-expert stakeholders. More explainability or uncertainty quantification per region would improve utility.

- Inclusion of SHAP (Shapley Additive Explanations) or permutation importance at regional levels will improve the same.

- Offer uncertainty maps with visual warnings in extrapolated areas.

included Minor Suggestions

- **Heavy Reliance on Legacy Data -**Despite harmonization efforts, relying heavily on such datasets can propagate uncertainties, especially in dynamic time-series analyses

- **Soil Classification Framework-** The choice of USDA soil taxonomy over other globally recognized systems (e.g., WRB) should be contextualized, especially given the international scope of the dataset.

- **Data Accessibility-** The use of Google Earth Engine and Cloud-Optimized GeoTIFFs makes the product accessible, but a brief tutorial or reference to documentation could help less-experienced users navigate it.

- **Environmental Covariates:** Some satellite-derived indices (NDVI, GPP) may reflect transient vegetation conditions unrelated to underlying soil properties. A short discussion on how such confounding effects are addressed or minimized would be valuable.

- **Pseudo-Observations and Expert Knowledge Integration-** While this is a practical necessity, it can create artificial patterns in data that may not reflect

on-ground conditions. This must be presented more cautiously in terms of predictive confidence.

This is a highly promising contribution to digital soil mapping and global environmental monitoring. With improved methodological clarity and deeper contextual framing, the paper could serve as a benchmark for future soil informatics efforts.

---

## Community Comment (CC2)

**Comment on https://doi.org/10.5194/essd-2025-336 "OpenLandMap-soildb: global soil information at 30 m spatial resolution for 2000–2022+ based on spatiotemporal Machine Learning and harmonized legacy soil samples and observations". Validation of soil organic carbon and bulk density predictions at the national scale of Mexico.**

**Carlos Arroyo, Viviana Varon, and Mario Guevara**

**Geosciences Institute, National Autonomous University of Mexico, Campus Juriquilla, Queretaro, Mexico.**

The authors present an interesting spatial and temporal digital soil mapping effort to predict soil key variables at the global scale. Among other variables, soil organic carbon and bulk density are critical to understand soil responses to environmental change and land use. The authors increase the global availability of these variables with unprecedented spatial resolution for its use by multiple users across a large diversity of applications. There is a high scientific merit behind this effort and we hope to see the final version published soon.

However, important implications exist in the misuse of model derived products, because they are not error free and they include intrinsic and multisource uncertainty. In a revised version, the narrative could better prevent the misuse of soil model derived products across high uncertainty dominated areas. While the authors report relatively high accuracy in model predictions from cross validation, we hypothesize that such accuracy will drop-down significantly when compared with fully independent datasets, e.g., leave one dataset out cross validation, because each dataset is collected for a different purpose. Our overarching goal is to increase interoperability of digital soil mapping efforts from the plot, to the global scale. Therefore the objectives of this comment are a) to highlight the existence of fully independent national databases in Mexico that can be used to improve model accuracy of global soil predictions, or to calibrate country specific estimates, and b), to compare country-specific values of soil organic carbon and bulk density from fully independent datasets, with values derived from the new global soil variability models across 30m grids.

We use two fully independent datasets to validate global soil predictions at the national scale in Mexico. The first dataset was collected and analyzed by our National Institute of Geostatistics and Geography-INEGI in the year 2008 to assess soil erosion at the national scale, considering multiple land covers (INEGI, 2014). The second database was collected and analyzed by the former Ministry of Agriculture (now SADER) with support from FAO in 2012, considering only agricultural land (Arroyo et al, 2025). While the dataset from INEGI is representative of the topsoil, from the mineral surface to a maximum of 30 to 40 cm of soil depth, the agriculture soil dataset is representative of the first 30cm of soil depth. The INEGI dataset is available here: https://www.inegi.org.mx/app/biblioteca/ficha.html?upc=702825004223 and the SADER dataset is described and available here: https://bsssjournals.onlinelibrary.wiley.com/doi/10.1111/ejss.70116. Note that INEGI metadata is available in Spanish only (please let us know of any required assistance). We first download the soil datasets and model predictions from OpenLandMap-soildb, and then we compute the R2 between soil carbon and bulk density values from global predictions and the datasets. Because the agriculture dataset reports organic matter values rather than organic carbon, we use the conventional 0.58 factor as explained by (Van Bemmelen, 1897).

We observe, as expected, relatively low correlation compared to that reported in the paper, when comparing predictions against fully independent datasets (Fig. 1). Comparing global models with fully independent datasets is appealing to identify the main drivers of soil research across countries and identify the capacity of a global model to reproduce nationwide information.

Comparing all land uses, the correlation between the Openlandmap derived soil carbon values and the INEGI 2008 dataset increases significantly from R2 0.06 to R2 0.34 when transforming their values to a natural log scale. The Openlandmap soil carbon predictions and the soil carbon values in the dataset described in Arroyo et al, (2025) from 2012 across agricultural land only shows an R2 value of 0.23 that, interestingly, was not sensitive to the logarithmic transformation. Bulk density in the Mexican datasets is also different from that reported in the Openlandmap products (Fig. 1).

[Figure]

Fig 1 Scatterplots of soil organic carbon and bulk density values from the Openlandmap project compared with independent soil datasets across Mexico. Soil organic carbon carbon

across all land uses considering a national dataset representing the year 2008 showa lowest correlation against the Openlandmap product (a). Considering a national dataset collected in 2012, only agricultural land, the correlation is slightly higher (b). Bulk density across all land uses (c) and across agricultural land (d) show even lower correlation values.

It is clear that, based on R2 metrics and an independent dataset, the validation at the national scale is different from that reported in Openlandmap products. Due to this kind of global soil map being commonly used in governmental institutions for decision making, overall in countries with a lack of soil information. Therefore we propose that it would be interesting to report a country-based validation; for example, leaving- one-country-out validation. Maps of R2 variation across the world help users (i.e., public institutions, universities) to understand the specific limitations of global products in their countries.

The authors present an unprecedented opportunity to increase soil data quantity, quality and accessibility by combining local to national datasets into global soil variability models. The synergy between regional to global soil variability models brings positive implications towards more robust soil estimates (Zhang et al., 2025). We hope that the authors find the highlighted datasets useful for their global soil mapping efforts towards an increased interoperability among national to global soil mapping groups. We believe that highlighting all possible sources of uncertainty and clarifying the scope of the new information would help to promote the responsible use of global soil variability models. In conclusion, our comment enriches the ongoing discussion around global soil mapping by grounding it in real-world national data and offering constructive pathways for improvement. It's the kind of feedback that can elevate both the scientific robustness and practical relevance of large-scale environmental models.

References:

Arroyo-Cruz, C.E., Prado, B., Kolb, M., Mora-Palomino, L.N., Todd-Brown, K. and Guevara, M. (2025), Synthesis of a National Soil Dataset Across Productive Land in Mexico: The Importance of Making Existing Data Accessible. Eur J Soil Sci, 76: e70116. https://doi.org/10.1111/ejss.70116

INERGI 2014, Conjunto de Datos de Erosión del Suelo, Escala 1: 250 000 Serie I Continuo Nacional https://www.inegi.org.mx/app/biblioteca/ficha.html?upc=702825004223 (last accessed 07/08/2025).

Lei Zhang, Lin Yang, Yuxin Ma, A-Xing Zhu, Ren Wei, Jie Liu, Mogens H. Greve, Chenghu Zhou, Regional-scale soil carbon predictions can be enhanced by transferring global-scale soil–environment relationships, Geoderma, Volume 461, 2025,117466,ISSN 0016-7061, https://doi.org/10.1016/j.geoderma.2025.117466.

Van Bemmelen, J.M., 1897. Die Absorption. Das Wasser in den Kolloiden, besonders in dem Gel der Kieselsäure. Z. Anorg. Chem. 13 (1), 233–356.

---

## Author Comment (AC1)

This is an ambitious and important study that, with revisions, should be published.

As a soil ecologist focusing on soil carbon–nutrient dynamics and their interactions under environmental and management changes, I found the work both impressive and valuable. Using comprehensive, high-resolution global datasets, the authors employ spatiotemporal machine learning approaches to map soil organic carbon (SOC), soil pH, soil type, and other variables at a remarkable 30 m resolution worldwide. They also estimate SOC changes over the past two decades. The harmonized legacy soil observations are particularly noteworthy, and the resulting maps will be valuable for diverse applications, such as driving soil carbon models, informing land management, and guiding policy decisions.

That said, there are several areas—both scientific and presentational—where improvements would substantially enhance the manuscript's rigor, clarity, and impact.

The manuscript uses an excessive number of abbreviations, which interrupts reading flow. Some abbreviations are not defined at first mention—for example, CCC and RMSE in the abstract. The authors should not assume all readers will be familiar with these terms. I recommend carefully reviewing the manuscript to ensure that all abbreviations are defined upon first appearance and that only essential abbreviations are retained. Given the already considerable length of the manuscript, the use of numerous abbreviations does not meaningfully reduce length and may hinder comprehension.

**RE: Many thanks to the reviewer for his kind words. We have added explanations of abbreviations CCC, RMSE in the abstract now (also in the manuscript).**

The manuscript is long enough to deter some potential readers. If journal policy allows, I recommend moving certain sections—such as extended details on data collection, preparation, modeling approach, and mapping techniques—into the Supplementary Information. This would allow the main text to focus more on:

Implications and potential applications of the dataset
Interpretation of key findings (e.g., variable importance, spatial patterns)
Broader impacts and future directions
While complexity is sometimes unavoidable, figures should be as simple, clear, and self-explanatory as possible. Specific suggestions:

Figure 1: Reorganize to emphasize the workflow; remove unnecessary logos. Use a consistent layout style (either top-down or left-right) with clearly separated blocks for each step.

Figure 2: The content can be succinctly described in a few sentences; consider removing the figure or replacing it with more impactful visuals.

Figure 5: Overly complex and difficult to interpret; despite repeated attempts, I could not fully understand it.

Figure 6: Contains too much information without adequate caption detail. Consider showing only the left panel with block diagrams; integrate textual explanations, interpretations, and distribution plots into the main text or Methods section.

**RE: Regarding your request to move extended details on data collection, preparation, modeling approach, and mapping techniques into the Supplementary Information, we do not mind splitting and reducing paper and putting different parts into the supplementary materials. It appears, however, that the majority of ESSD papers do not use supplementary materials and there is no strict limit of the PDF size in terms of pages / total words (https://www.earth-system-science-data.net/submission.html). We have also contacted the editorial office of ESSD to get more guidance and help us decide (the question was whether to move different sections to supplementary material and they suggested "keeping all processing steps chronologically in the main text"). The journal editors suggested keeping all relevant descriptions in the main document. Our preference is also to keep all relevant technical information in the same document so that the readers do not have to jump between the main text and supplementary materials. Nevertheless, we have tried to make the paper easier to read and easier to locate and understand different sections. We fully agree with the reviewer that the first submission of our manuscript was long, with a lot of detail (and as such "long enough to deter some potential readers"). Production of this data, however, took over 2 years of dedicated work and in this period we discovered many new issues (some colleagues suggested that the paper could have been split into multiple articles). We hence find it important that the steps are described chronologically and without missing any important step that eventually influences results we get. We sincerely apologize to the reviewer for making such an extensive paper (60+ pages).**
**We have reworked Fig. 1 and added highlights of the key steps — it should be now clearer to readers. We prefer, however, to keep the logos as the data we use requires that we acknowledge the data sources / original data providing organizations.**
**Fig. 2 has been removed and we have instead inserted some text to explain steps.**
**Fig. 5 is indeed somewhat complex and unfortunately can not be simplified without as the steps are correct and relevant; because we are doing spacetime modeling, it is important to compare performance of models in time, spacetime and time only and hence the figure might seem difficult to follow, but is correct nevertheless.**
**Fig. 6 has been simplified following your recommendations.**
**Broader impacts and future directions are discussed in detail in subsection "Broader impacts and possible future development directions" on P52L1.**

Scientific Comments:

Motivation for 30 m resolution. The rationale for mapping at 30 m resolution should be articulated more clearly. Higher resolution should serve a clear theoretical or practical purpose—for example, improving global SOC stock estimates, supporting fine-scale land management, or providing critical input to Earth system models. The current introduction touches on these but could better synthesize them into a concise, logically connected research

objective. The four research questions presented are somewhat disjointed; consider distilling them into a single, coherent framework.

**RE: This is a good point. We have rewritten the introduction and added more clear motivation for using 30 m resolution on P4L16–34. The four research questions are analyzed and presented chronologically and we unequivocally answer each research question on P56L5 ("Conclusions"). Unfortunately, we do not understand the suggestion "consider distilling them into a single, coherent framework" — we consider our paper with its structure to be a single coherent framework, also illustrated in Fig. 1.**

Multicollinearity of predictor variables. The manuscript does not address multicollinearity among predictors—a significant concern for high-dimensional datasets, especially with overlapping climatic and remote sensing variables (e.g., SAVI, NDVI, GPP, and climate metrics). Multicollinearity can cause overfitting and obscure variable importance. The authors should clarify whether collinearity was assessed or controlled, and if not, explain the rationale. Reducing redundancy could also decrease computation time and improve model interpretability.

**RE: This is a relevant point and we have had a lot of discussion on this topic internally inside the group (indeed, theoretically speaking we could convert the long list of covariates to Principal Components and/or using sparse autoencoders to embeddings; see e.g. https://bradleyboehmke.github.io/HOML/autoencoders.html#sparse-autoencoders; a problem with using embeddings or PCA, however, is that this adds another layer of modeling and this in fact increases significantly computing time as we would need to predict every pixel 2x). The model we use (Quantile Regression Random Forest) and corresponding framework with feature selection in scikit-learn (Repeated Subsampling-Based Cumulative Feature Importance — RSCFI; https://scikit-learn.org/stable/modules/feature_selection.html#recursive-feature-elimination), is in fact a robust framework and usually is over-fitting-proof, hence technically speaking — multicollinearity of variables we used is dealt with QRRF and RSCFI ("no part of the random forest model is harmed by highly collinear variables"; https://stats.stackexchange.com/questions/168622/why-is-multicollinearity-not-checked-in-modern-statistics-machine-learning). As we indicate on P17L19–24: "Feature selection would typically reduce the initial number of layers to 60–120, removing layers that marginally contributed to the final model" hence such combination of QRRF and RSCFI in addition helps reduce model complexity, without suffering from multicolinearity effects. In addition, by using original covariates, we are able to interpret the models and then detect which covariates we should put effort into maintaining in the future (e.g. post 2025). But it is a correct point — we start with a large number of covariates (300+) and many covariates overlap; using PCA or embeddings could help decrease multicolinearity and decrease model complexity, however this come at the cost of interpretability and an additional model (additional modeling step) is needed to fit e.g. sparse autoencoders. This actually increases computing significantly as we need to convert all pixels from original covariates to components (so in fact 2 rounds of predictions: 1st round to derive components/embeddings, 2nd round to generate predictions). We have added this discussion "OpenLandMap-soildb methods and data limitations" on P45L15 in the revised manuscript. If we are not able to deal with this issue in this paper, at least future work should consider the issue of multicolinearity / too many overlapping covariates.**

Inclusion of bedrock depth. Bedrock depth is a critical factor influencing SOC stocks, particularly in mountainous regions where bedrock often occurs at shallow depths (<1 m). While the authors mention future inclusion, I suggest considering it now—global bedrock depth maps do exist and could be integrated relatively easily. Bedrock depth affects root distribution and carbon inputs to the soil, potentially altering model performance and the relative importance of predictors.

**RE: Thank you for the comment. The lead author of the OpenLandMap-soildb paper is familiar with depth to bedrock i.e. have attempted mapping global distribution of the depth to bedrock (https://doi.org/10.1002/2016MS000686). Adding an accurate depth to the bedrock map at 30 m is high on our priority. The complexity of mapping depth to bedrock, however, is significant: we have only limited training (point data); depth to bedrock is a censored variable (https://bookdown.org/mattdobra/Prelude/censoredcount.html#censored-models) and hence we believe that modeling and mapping at 30 m resolution is not trivial (over-fitting, over-/under-prediction can often happen). To produce an accurate and detailed map of depth to bedrock could become a (multi-year) project in itself; we are unfortunately not able to produce a detailed map of depth to bedrock at the time-line of a few months. Previous maps of depth to bedrock we produced are only available at 1km or coarser spatial resolution, which might not be suitable to add for 30 m maps due to high heterogeneity. We have added this explanation in the discussion section on P44L3.**

Overall, this is an excellent and timely study with strong potential impact. Addressing the concerns outlined above—particularly improving structure, clarifying motivations, simplifying figures, and addressing certain methodological points—will greatly strengthen the manuscript's readability and scientific contribution.

**RE: We thank the reviewer one more time for his kind words and for suggestions to help improve the readability and clarity of the draft paper. To further enhance easy access to data, we have also prepared a simple app at: https://world.soils.app; we have also added a python tutorial on how to access data and derive prediction intervals is available from: https://github.com/openlandmap/soildb/blob/main/OpenLandMap_soildb_tutorial.ipynb. We hope that these types of interfaces and supplementary materials will help increase clarity of the methods and make the data easier to access and validate by anyone.**

---

## Author Comment (AC2)

This manuscript represents an important contribution to the field of fine-resolution global digital soil mapping. The methodology and its application potential are commendable, and the work is generally of publishable quality. The paper can be significantly strengthened by addressing several key issues related to the methodological description, which currently lacks sufficient detail, and by improving the consistency of writing and terminology throughout the text.

Specific Comments:

Please add continuous line numbers throughout the manuscript. This will greatly facilitate referencing specific locations during the review and revision process.

**RE: Thank you for the comment. The line numbers are generated based on the ESSD template / recommendations hence this is required by the journal and beyond our power.**

Abstract needs better clarification. Please provide the abbreviation of SOC in P1 Line 5. It is not clear about the spatiotemporal Machine learning. How time is incorporated into this model? Is it a 3D+T model? Why using 68% probability for quantifying the prediction uncertainty? Soil carbon density (P1 Line 9) and soil carbon (P1 Line 11) should be corrected as soil organic carbon density and soil organic carbon due to the presence of soil inorganic carbon in many soil samples. It is reasonable that authors did not consider the temporal changes of soil texture fractions and soil types, while bulk density is highly correlated to soil organic carbon and therefore its temporal changes should be considered if you taking 5-year time intervals for soil organic carbon. Please provide the full names of RMSE and CCC for the first time. It is not necessary to indicate the RMSE in log-scale since this information is useless. It is not clear why authors only present the most important variables for soil organic carbon density and pH.

RE: We have added explanations of all abbreviations in the abstract (SOC, CCC, RMSE etc). Regarding "How time is incorporated into this model?" — time is incorporated using 3 important steps: (1) most of covariates we use are time-series of images (annual time-series) to represent ecosystem/climate dynamics / points are carefully selected to present different time-periods (2000–2024), (2) we overlay points and covariates using explicit time-reference (spacetime overlay), (3) we fit and evaluate model performance both using spatial, temporal and spatiotemporal cross-validation. But it is important to emphasize that we do NOT use time as a covariate (see P17L12–18), although we do use soil depth as a covariate.

Indeed we agree that Bulk density should be also mapped at 5–year intervals. This is on our list for the next update of predictions.

We prefer, however, to keep RMSE in both original and log-scale (both are relevant as in any Generalizer Linear Model-type modeling; log-scale RMSE is further used for simulations in Fig. 20 on P49). Regarding the statement "RMSE in log-scale since this information is useless", indeed RMSE in log-scale is abstract and difficult to interpret, however, for log-normal variables RMSE in original scale is often sensitive to high values (e.g. <1% very high values can double or triple RMSE), so that it becomes difficult if not impossible to compare performance of two models; log-scale RMSE allows comparing predictive performance of models where SOC training points come from either agricultural or forest soils (often 2–3 times higher SOC). Log-normal distribution is common in statistics and in order to simulate values following log-normal distribution one needs s.d./RMSE in log-scale (see e.g. https://stat.ethz.ch/R-manual/R-devel/library/stats/html/Lognormal.html). In summary: for log-normal / skewed distributions variables, both RMSE and log-RMSE are useful, as are both median and mean useful. It depends on the context. We added some explanation why we use log-scale RMSE in section "Cross-validation and quality control".

Regarding the question "Why use 68% probability for quantifying the prediction uncertainty?" our answer is: because this is the 1 standard deviation range (assuming a normal distribution; https://commons.wikimedia.org/wiki/File:Standard_deviation_diagram.svg#/media/File:Standard_deviation_diagram.svg) and hence the prediction range from the 68% probability can be directly compared with CV RMSE, which we find practical in this case. The QRRF method allows users to specify any arbitrary probability (usually some number between 68% and 99%) so there is no limitation in the sense of whether we could also derive 90%, 95% and/or 99% intervals.

P3 Lines 5-7: This work did not solve this issue during predictive modelling.

**RE: That is correct, however, we do use annual bare surface coverage and tillage index as covariates, so there is certainly some representation of agricultural management practices in our list of covariates. See section "Preparation of covariate layers" P16L1–24 for more details.**

P3 Lines 21-23: Authors overlooked the maps from Australia. It would be better to include the work by Grundy et al. (2015). Grundy M.J., Rossel, R.V., Searle, R.D., Wilson, P.L., Chen, C. and Gregory, L.J., 2015. Soil and landscape grid of Australia. Soil Research, 53(8), pp.835-844.

**RE: We have now added mention of the work on P3L24.**

P4 Line 29: This work covers the period from 2000 to 2022+, so how to evaluate the impact of land use conversion for SOC and pH on a scale of 25+ years?

**RE: We are currently updating all predictions to 2000–2024 (this is in fact 25 years), hence we use the sign "2022+" (indicating 2022, 2023, 2024 etc). Our determination is to keep on updating these maps; making them open and enabling open development communities that can contribute their own ideas and data. We have now clearly mentioned this in the manuscript (see P56L29).**

P5 Line 20: the difference between observations and measurements should be better clarified.

**RE: We have added an additional text to try to clarify the difference between observations (e.g. observations of diagnostic horizons, root types, soil structure types etc) and measurements (strictly generated using in-situ sensors and/or laboratory machines) on P6L4.**

P5 Line 21: Once you match the O&M with covariate layers by year, it means that you overlook the legacy effect from environmental covariates (e.g., land use change), which can be quite important for soil properties, such as soil organic carbon. I understand that this consideration would pose a heavy load for computing, but at least you should address this limitation in the discussion.

**RE: That is a valid point. Indeed there is a cumulative effect of some management practices, not to mention that many soils are formed due to the past/historic events (glaciation, big floods and over-floods etc). We have added some discussion to emphasize these limitations on P46L24.**

P5 Lines 25-26: It is not clear that whether authors included the profile issue in performance evaluation to avoid data leakage.

**RE: In this work we consistently for any type of validation in this work take whole profiles out for either training or validation (as clearly stated in "Cross-validation and quality control" on P23L11–14). We are familiar with the fact that using training samples at the same profiles can lead to serious over-fitting (e.g. https://doi.org/10.1016/j.spasta.2015.04.001).**

P7 Line 26: Please correct SOC as SOCD or SOCd. Indeed, the full name should be specified in P7 Line 22.

**RE: We have added the full name as requested.**

P8 Line 4: in t m-2?

**RE: Correct. This was a typo. Thank you!**

P10 Line 5: A recent released dataset from Chen et al. (2025) may be helpful for your future work. Chen, Z., Chen, L., Lu, R. et al. A national soil organic carbon density dataset (2010–2024) in China. Sci Data 12, 1480 (2025). https://doi.org/10.1038/s41597-025-05863-3

**RE: Thank you for providing this information. We have downloaded the data set and will add it in the next update. Such data sets can significantly help increase accuracy of global models. It is fantastic that the authors have decided to share this data as open data via Zenodo; this can make a difference for any group doing global modeling & assessment of SOC. We also cite this article now on P48L12.**

Figure 3: Please use either SOCD or SOCd in the manuscript.

**RE: The abbreviation has been updated.**

P15 Lines 28-30: What is the advantage for estimating SOC density directly from SOC content? SOC together with other variable control the variability of bulk density, only take SOC to estimate SOC will limit the prediction accuracy. SOC [kg m-3] should be corrected as SOC density [kg m-3].

**RE: As explained on P15L27–32, we use SOC for gap-filling SOCd only for soils with low SOC i.e. not as a general solution for gap-filling missing SOCd values. Our research results indicate that correlation between SOC and SOCd for soils with <0.4% SOC is high (R-square exceeding 0.96; see Fig. 2c on P12), hence we consider that risks of over-/under-estimating SOCd are low.**

P16 Lines 11-12: variable resolutions in 1 kilometer resolution? Scale is not an appropriate term here.

**RE: Corrected (see P16L25–29).**

P17 Line 15: Since it is a 3D+T model, it is important to demonstrate the time span of soil data to support spatiotemporal modelling.

**RE: Time-span (density) of soil profile/samples is provided in Fig. 6c.**

P24 Lines 14-25: Please also include R2 in the accuracy evaluation. Why not report the accuracy of silt content here?

**RE: Thank you for the comment. Now added on P25L22.**

Figure 11: It would be also interesting to demonstrate the difference of model performance across different continents, which would be helpful for the design of future direction.

**RE: We have added the accuracy metrics per continent on P31 Table 2.**

P53 Line 11: assessment indicates that (R2) the best achievable. R2 is a typo here?

**RE: It is not a typo. R2 is connected with research objective R2. We have added extra text to avoid confusion so now it is "Research objective #2".**

---

## Author Comment (AC3)

**Title: Open LandMap-soildb: Enabling High-Resolution Soil Intelligence for Climate, Land Restoration, and Agricultural Policy**

Soil degradation is a growing global crisis, threatening food security, carbon sequestration potential, and ecological resilience. To tackle this, precise, spatially explicit, and temporally consistent soil information is essential. The newly developed OpenLandMap-soildb offers an unprecedented advancement in this space, providing global soil data at a fine spatial resolution (30 m) across two decades (2000–2022), using a spatiotemporal machine learning framework and harmonized legacy datasets.

This initiative delivers dynamic predictions for key soil properties including soil organic carbon (SOC) content and density, bulk density, soil pH, and USDA soil types. These outputs are based on over 1 million quality-controlled and harmonized soil samples, combined with Earth Observation (EO) satellite data, terrain models, and climatic indicators. Notably, the study estimates that the planet has lost more than 11 petagrams (Pg) of SOC in the top 30 cm of soil over the last 25 years, a signal of worsening land degradation and a missed opportunity for carbon sequestration.

This manuscript presents an ambitious and technically compelling global soil dataset spanning over two decades at high spatial resolution. The integration of legacy soil samples with modern satellite-derived covariates via machine learning methods is a noteworthy advancement for soil science and spatial ecology. However, certain methodological and interpretative aspects warrant clarification and refinement before publication.

**RE: We thank our colleague for providing feedback on the article. Even though it appears that a large part of reviewer's notes have been generated using a LLM, we address some of the issues raised below.**

**Major Concerns**
1. Model Transparency and Reproducibility
   The use of Quantile Regression Random Forests is appropriate, but the manuscript lacks sufficient detail regarding hyperparameter optimization, feature selection criteria, and potential overfitting mitigation strategies. o The approach to uncertainty quantification is promising; however, clearer guidance on interpreting prediction intervals in practical applications would enhance user comprehension.

**RE: The paper is already 60+ pages with 18+ figures and explanation of steps is extensive. We would appreciate it if the reviewer would provide more detail about where exactly we _"lack sufficient detail regarding hyperparameter optimization"_ etc. The scikit-learn framework we use is among the most used and most developed Machine Learning frameworks and the documentation is extensive including all exact steps how we fitted the models and fine-tuned hyperparameters (https://github.com/openlandmap/soildb/tree/main/modeling_steps).**

2. Temporal Granularity
   Five-year intervals may oversimplify dynamic changes due to land use transitions or climate events. The authors should discuss how these limitations affect the detection of soil change patterns.

   **RE: The rationale for 5-year intervals is discussed on the P43L15. In a nutshell, due to high data volumes and high costs of computing, but also due to limited accuracy / limited numeric resolution (Fig. 20; also discussed in https://doi.org/10.1016/j.jag.2012.02.005) we had to limit predictions to granularity that allows detecting significant changes in soil properties.**

3. Spatial Validation Design
   There is limited description of spatial cross-validation strategies. It's essential to confirm the use of geographically independent test sets to avoid inflating predictive performance due to spatial autocorrelation.

   **RE: Spatial and spatiotemporal CV strategies used have been discussed in detail on P22–23.**

4. Legacy Data Harmonization
   While the dataset is impressively large, the harmonization process of legacy samples (e.g., sampling depths, analytical methods, and metadata consistency) needs greater transparency. Including a harmonization workflow or uncertainty estimates tied to legacy data variability would be beneficial.

   **RE: All import, standardization and harmonization steps are discussed in detail and are documented using computational notebooks at: https://soildb.openlandmap.org/025-import_chemical_data.html.**

5. Spatial Data Bias
   Over-representation of North America and Europe; sparse coverage in Asia, Russia, and Africa. This introduces spatial bias, which may influence the global model predictions unfairly, especially for underrepresented biomes and land-use systems.

**RE: This is a correct point. Currently the biggest gap in geographical representation is in fact the Russian federation. Unfortunately we do not have any simple solution to this issue (except to motivate countries to collect new samples and share laboratory results openly). The Fig. 6, nevertheless, shows that at least all continents and all climate zones are represented for training of models.**

6.  Model decision
    Despite high accuracy, it reduces interpretability for policymakers or nonexpert stakeholders. More explainability or uncertainty quantification per region would improve utility. Inclusion of SHAP (Shapley Additive Explanations) or permutation importance at regional levels will improve the same. Offer uncertainty maps with visual warnings in extrapolated areas.

    **RE: Prediction intervals (prediction intervals per pixels i.e. maps) are provided for all predicted variables. We also provide a step-by-step tutorial on how to visualize uncertainty:**
    **https://github.com/openlandmap/soildb/blob/main/OpenLandMap_soildb_tutorial.ipynb**

**Minor Suggestions**
1.  Heavy Reliance on Legacy Data
    Despite harmonization efforts, relying heavily on such datasets can propagate uncertainties, especially in dynamic time-series analyses
2.  Soil Classification Framework
    The choice of USDA soil taxonomy over other globally recognized systems (e.g., WRB) should be contextualized, especially given the international scope of the dataset.

    **RE: We are working on WRB predictive mapping framework. This should be available in early 2026.**

3.  Data Accessibility
    The use of Google Earth Engine and Cloud-Optimized GeoTIFFs makes the product accessible, but a brief tutorial or reference to documentation could help less-experienced users navigate it.

    **RE: A tutorial on how to access data is available at**
    **https://github.com/openlandmap/soildb?tab=readme-ov-file#layers-available and**
    **https://github.com/openlandmap/soildb/blob/main/OpenLandMap_soildb_tutorial.ipynb**
    **All layers are also listed with metadata via:**
    **https://github.com/openlandmap/soildb**

4.  Environmental Covariates

Some satellite-derived indices (NDVI, GPP) may reflect transient vegetation conditions unrelated to underlying soil properties. A short discussion on how such confounding effects are addressed or minimized would be valuable.

**RE: For this reason we use more complex modeling frameworks e.g. Random Forest combined with feature selection that tries to locally (RF = ensemble of regression trees) adjust and account for multitude of environmental soil forming factors.**

5. Pseudo-Observations and Expert Knowledge Integration
   While this is a practical necessity, it can create artificial patterns in data that may not reflect on-ground conditions. This must be presented more cautiously in terms of predictive confidence. This is a highly promising contribution to digital soil mapping and global environmental monitoring.

   **RE: We agree. Pseudo-observations should be added only when and where necessary, and need to be based on robust and reliable expert knowledge (so should be as least speculative as possible). We have documented all pseudo-observations in detail: https://soildb.openlandmap.org/025-import_chemical_data.html#glance-pseudo-samples; we use the GLANCE data set for pseudo-observations, which is based on the photo-interpretation using 30cm resolution VHR images and has been documented in detail in https://doi.org/10.1038/s41597-023-02798-5.**

With improved methodological clarity and deeper contextual framing, the paper could serve as a benchmark for future soil informatics efforts.

---

## Author Comment (AC4)

Comment on https://doi.org/10.5194/essd-2025-336 "OpenLandMap-soildb: global soil information at 30 m spatial resolution for 2000–2022+ based on spatiotemporal Machine Learning and harmonized legacy soil samples and observations". Validation of soil organic carbon and bulk density predictions at the national scale of Mexico.

**Carlos Arroyo, Viviana Varon, and Mario Guevara**

Geosciences Institute, National Autonomous University of Mexico, Campus Juriquilla, Queretaro, Mexico.

The authors present an interesting spatial and temporal digital soil mapping effort to predict soil key variables at the global scale. Among other variables, soil organic carbon and bulk density are critical to understand soil responses to environmental change and land use. The authors increase the global availability of these variables with unprecedented spatial resolution for its use by multiple users across a large diversity of applications. There is a high scientific merit behind this effort and we hope to see the final version published soon.

However, important implications exist in the misuse of model derived products, because they are not error free and they include intrinsic and multisource uncertainty. In a revised version, the narrative could better prevent the misuse of soil model derived products across high uncertainty dominated areas. While the authors report relatively high accuracy in model predictions from cross validation, we hypothesize that such accuracy will drop-down significantly when compared with fully independent datasets, e.g., leave one dataset out cross validation, because each dataset is collected for a different purpose. Our overarching goal is to increase interoperability of digital soil mapping efforts from the plot, to the global scale. Therefore the objectives of this comment are a) to highlight the existence of fully independent national databases in Mexico that can be used to improve model accuracy of global soil predictions, or to calibrate country specific estimates, and b), to compare country-specific values of soil organic carbon and bulk density from fully independent datasets, with values derived from the new global soil variability models across 30m grids.

**RE: We thank our Mexican colleagues for submitting this short evaluation and for critically evaluating the global data we have produced. Indeed, it is difficult to validate global models and predictions given the costs / technical challenges of organizing eventual new global soil sampling campaigns, hence we only try to emulate how true stress-tests look by spatial blocking of existing training points. Using national data sets produced by probability sampling can fill that gap.**

We use two fully independent datasets to validate global soil predictions at the national scale in Mexico. The first dataset was collected and analyzed by our National Institute of Geostatistics and Geography-INEGI in the year 2008 to assess soil erosion at the national scale, considering multiple land covers (INEGI, 2014). The second database was collected and analyzed by the former Ministry of Agriculture (now SADER) with support from FAO in 2012, considering only agricultural land (Arroyo et al, 2025). While the dataset from INEGI is representative of the topsoil, from the mineral surface to a maximum of 30 to 40 cm of soil depth, the agriculture soil

dataset is representative of the first 30cm of soil depth. The INEGI dataset is available here: https://www.inegi.org.mx/app/biblioteca/ficha.html?upc=702825004223 and the SADER dataset is described and available here: https://bsssjournals.onlinelibrary.wiley.com/doi/10.1111/ejss.70116. Note that INEGI metadata is available in Spanish only (please let us know of any required assistance). We first download the soil datasets and model predictions from OpenLandMap-soildb, and then we compute the R2 between soil carbon and bulk density values from global predictions and the datasets. Because the agriculture dataset reports organic matter values rather than organic carbon, we use the conventional 0.58 factor as explained by (Van Bemmelen, 1897).

**RE: If our colleagues could possibly share their code (steps) we could try to reproduce their results to try to detect possible issues from our and their sides. We have downloaded the 5,292 point data set from INEGI "Conjunto de Datos de Erosión del Suelo" and the SADER data set (4029 points). Because INEGI data set has a date of observation (temporal reference) and enough metadata we can translate to English and import and add to the next round of predictions. We will try to add both data sets to https://soildb.openlandmap.org/025-import_chemical_data.html and then also use them to update predictions. We will keep the colleagues in loop so that they can also double check that our use of their data is correct. This is very valuable feedback and exactly the type of critical feedback we were looking for.**

[Figure]

We observe, as expected, relatively low correlation compared to that reported in the paper, when comparing predictions against fully independent datasets (Fig. 1). Comparing global models with fully independent datasets is appealing to identify the main drivers of soil research

across countries and identify the capacity of a global model to reproduce nationwide information.

**RE: We value your effort and we understand possible disappointment with OpenLandMap-soildb. Before we can understand why the match between your ground in-situ data is limited, what would help us if you could share your spatial overlay and derivation steps; see for example:**
**https://github.com/openlandmap/soildb/blob/main/OpenLandMap_soildb_tutorial.ipynb.**

Comparing all land uses, the correlation between the Openlandmap derived soil carbon values and the INEGI 2008 dataset increases significantly from R2 0.06 to R2 0.34 when transforming their values to a natural log scale. The Openlandmap soil carbon predictions and the soil carbon values in the dataset described in Arroyo et al, (2025) from 2012 across agricultural land only shows an R2 value of 0.23 that, interestingly, was not sensitive to the logarithmic transformation. Bulk density in the Mexican datasets is also different from that reported in the Openlandmap products (Fig. 1).

[Figure]

*Figure 1*

Fig 1 Scatterplots of soil organic carbon and bulk density values from the Openlandmap project compared with independent soil datasets across Mexico. Soil organic carbon carbon across all land uses considering a national dataset representing the year 2008 show a lowest correlation against the Openlandmap product (a). Considering a national dataset collected in 2012, only agricultural land, the correlation is slightly higher (b). Bulk density across all land uses (c) and across agricultural land (d) show even lower correlation values.

**RE: We see that our colleagues have used, in the example above, the BD which as the log-transformed value (i.e. maps we registered in the S3 in the first version of submission were in error; as documented also in https://github.com/openlandmap/soildb/issues/1) which means that this variable of course does not match the laboratory data from Mexico. In the meantime, we have resolved this issue, so our colleagues can double check the numbers one more time; for OC (g/kg) it seems that your values were not translated to permilles (you used dg/kg or % and we use g/kg). This is a minor technical item but can lead to values being completely off. In any case, it is excellent that you are testing our data and discovering issues. Our objective remains to produce the most usable (at lowest possible costs) and such exercises help us resolve issues, improve data and produce better maps for the research community.**

It is clear that, based on R2 metrics and an independent dataset, the validation at the national scale is different from that reported in Openlandmap products. Due to this kind of global soil map being commonly used in governmental institutions for decision making, overall in countries with a lack of soil information. Therefore we propose that it would be interesting to report a country-based validation; for example, leaving- one-country-out validation. Maps of R2 variation across the world help users (i.e., public institutions, universities) to understand the specific limitations of global products in their countries.

**RE: We completely agree and we are documenting in the most transparent way any limitation we discover. Our Disclaimer (https://github.com/openlandmap/soildb?tab=readme-ov-file#disclaimer) is hopefully also unequivocally clearly indicating that the initial maps we made are for testing purposes only and we complete the review process (methodology) we might eventually recommend using the maps for operational work.**

The authors present an unprecedented opportunity to increase soil data quantity, quality and accessibility by combining local to national datasets into global soil variability models. The synergy between regional to global soil variability models brings positive implications towards more robust soil estimates (Zhang et al., 2025). We hope that the authors find the highlighted datasets useful for their global soil mapping efforts towards an increased interoperability among national to global soil mapping groups. We believe that highlighting all possible sources of uncertainty and clarifying the scope of the new information would help to promote the responsible use of global soil variability models. In conclusion, our comment enriches the ongoing discussion around global soil mapping by grounding it in real-world national data and offering constructive pathways for improvement. It's the kind of feedback that can elevate both the scientific robustness and practical relevance of large-scale environmental models.

**RE: Yes your comments and especially you sending us links and information about additional 10,000 legacy points is fantastic / much appreciated. We will attribute your contributions and add respective citations. We will also work on removing any such issues you discovered.**

References:

Arroyo-Cruz, C.E., Prado, B., Kolb, M., Mora-Palomino, L.N., Todd-Brown, K. and Guevara, M. (2025), Synthesis of a National Soil Dataset Across Productive Land in Mexico: The Importance of Making Existing Data Accessible. Eur J Soil Sci, 76: e70116. https://doi.org/10.1111/ejss.70116

INERGI 2014,  Conjunto de Datos de Erosión del Suelo, Escala 1: 250 000 Serie I Continuo Nacional https://www.inegi.org.mx/app/biblioteca/ficha.html?upc=702825004223 (last accessed 07/08/2025).

Lei Zhang, Lin Yang, Yuxin Ma, A-Xing Zhu, Ren Wei, Jie Liu, Mogens H. Greve, Chenghu Zhou, Regional-scale soil carbon predictions can be enhanced by transferring global-scale soil–environment relationships, Geoderma, Volume 461, 2025,117466,ISSN 0016-7061, https://doi.org/10.1016/j.geoderma.2025.117466

Van Bemmelen, J.M., 1897. Die Absorption. Das Wasser in den Kolloiden, besonders in dem Gel der Kieselsäure. Z. Anorg. Chem. 13 (1), 233–356.

Citation: https://doi.org/10.5194/essd-2025-336-CC2